# HI2M: Hard Inter- and Intra-Sample Masking for Dynamic Facial Expression Recognition

## Abstract

Dynamic facial expression recognition (DFER) holds significant potential for real-world applications. While existing methods have achieved promising results, they face two key limitations: (1) dependence on limited labeled data, and (2) equal treatment of all samples and regions during reconstruction, often resulting in sub-optimal attention to informative features. Recent masked autoencoder adaptations address the first limitation but inherit the second. To overcome these challenges, we propose Hard Inter- and Intra-sample Masking (HI2M), a novel framework comprising two synergistic components: Hard Sample Mining (HSM) and Hard Region Discovery (HRD). HI2M operates through a dual-masking strategy: First, HSM employs dynamic sample weighting based on inter-sample reconstruction loss deviations, automatically prioritizing challenging cases. Second, HRD utilizes a trainable reinforcement learning mechanism to identify and mask informative spatiotemporal regions by analyzing intra-sample variances, adaptively selecting varying numbers of regions per frame for fine-grained feature localization. This hierarchical approach-from sample-level importance weighting to region-level adaptive masking-enables focused learning on semantically rich facial dynamics while suppressing noise. Our comprehensive experiments on benchmark datasets demonstrate that HI2M significantly outperforms existing approaches, establishing new state-of-the-art performance in DFER.

## 1 Introduction

The field of facial expression recognition (FER) has gained significant research interest due to its critical applications in developing intelligent systems capable of empathetic human-machine interaction Pantic & Rothkrantz (2000); de Melo et al. (2023); Azazi et al. (2015); Huang et al. (2022); Whitehill et al. (2014). Conventionally, FER approaches are categorized into: (1) static FER (SFER) that analyzes expressions from individual images, and (2) dynamic FER (DFER) that examines temporal patterns in video sequences to model expression evolution. Since facial expressions naturally unfold over time, DFER has emerged as a particularly active research focus, offering more nuanced understanding compared to static approaches.

The field of DFER has seen the development of diverse deep learning architectures, primarily falling into three categories: (1) hybrid 2D CNN-RNN models Fan et al. (2016); Liu et al. (2018); Wang et al. (2020), (2) 3D CNN-based approaches Wang et al. (2023); Jiang et al. (2020); Liu et al. (2021); Wang et al. (2022a), and (3) Transformer frameworks Zhao & Liu (2021); Wang et al. (2022b); Ma et al. (2022); Li et al. (2023). While supervised methods have shown promising results, their performance is fundamentally constrained by the limited scale of annotated DFER datasets. While CLIP-based approaches Zhao & Patras (2023); Foteinopoulou & Patras (2024); Li et al. (2024a); Mai et al. (2024); Li et al. (2024b) have shown impressive transfer learning performance, they are limited by their inability to capture the nuanced temporal dynamics and spatial evolution characteristic of facial expressions. Recent work has turned to self-supervised learning paradigms, leveraging abundant unlabeled facial videos online. For example, MAE-DFER Sun et al. (2023) employs masked reconstruction to jointly encode facial appearance and temporal dynamics. HiCMAE Sun et al. (2024) integrates masked modeling with contrastive learning for multi-level feature enhancement. AVF-MAE++ Wu et al. (2025) introduces an adaptive dual-masking strategy across audio-visual modalities, creating a efficient cross-modal self-supervised objective.

However, our analysis reveals two problematic assumptions in current approaches: **(1) Equal sample importance**, where existing methods uniformly treat all samples despite evident differences in learning difficulty—as demonstrated in Figure 1(a), samples with high-amplitude facial variations (being more challenging) often receive inadequate attention, causing the model to overlook critical expression cues. **(2) Equal region importance**, where random or fixed masking strategies (Figure 1(b)) frequently select low-information regions for reconstruction, thereby failing to capture essential dynamic features and subtle expression transitions. These limitations collectively hinder models from effectively learning discriminative spatiotemporal patterns in facial expressions.

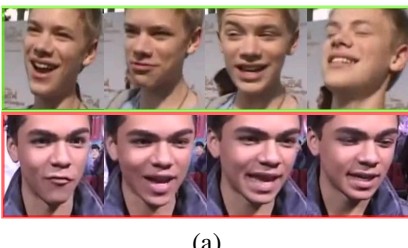 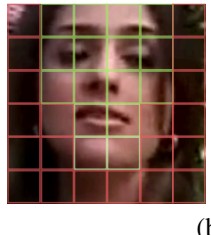 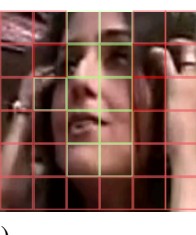

(a) (b)

Figure 1: Visualizations reveal two key aspects: **(a) High inter-sample differences** demonstrate that samples with greater expression variability (green boxes) present more challenging reconstruction targets compared to those with low variation (red boxes), underscoring the necessity of adaptive inter-sample modeling. **(b) Significant intra-sample variations** emerge when comparing frames within the same sample, where green-highlighted regions contain more discriminative information than red-marked areas, revealing the need of an adaptive region selection.

To address the aforementioned issues, we propose Hard Inter- and Intra-sample Masking (HI2M) for DFER, which comprises two key components: Hard Sample Mining (HSM) and Hard Region Discovery (HRD). HI2M optimizes MAE models with adaptive inter- and intra-sample masking that select important samples and regions for reconstruction. Specifically, pre-training datasets suffer from severe imbalance, where numerous easy samples overwhelm scarce yet valuable hard cases. While prioritizing challenging instances would boost model performance, self-supervised hardness assessment proves particularly difficult without explicit label guidance. The HSM is proposed to address this issue, which dynamically adjusts sample weights during training to prioritize difficult cases—particularly those containing complex spatiotemporal patterns. By quantifying inter-sample reconstruction difficulty, HSM automatically identifies and emphasizes hard-to-reconstruct samples near decision boundaries, ensuring the model allocates greater attention to semantically challenging instances. Furthermore, existing masking strategies fail to adequately prioritize complex yet discriminative regions, severely constraining models' capacity to capture essential spatiotemporal features. To address this limitation, we introduce HRD, which actively identifies and emphasizes regions containing critical dynamic information during reconstruction. Unlike conventional random masking approaches, HRD employs a reinforcement learning framework with policy gradient optimization, strategically rewarding the selection of challenging but informative intra-sample regions. This adaptive mechanism ensures the model concentrates its learning effort on high-value facial dynamics while suppressing less relevant areas. Extensive experiments on two benchmark datasets demonstrate the effectiveness of the proposed model. Our contributions can be summarized as follows:

- The proposed HI2M is the first unified framework that concurrently addresses hardness-aware masking at both sample and region granularities, overcoming the limitations of conventional single-level masking strategies in existing MAE variants.

- We propose Hard Sample Mining (HSM), an adaptive weighting module that automatically identifies and prioritizes hard-to-reconstruct samples through inter-sample difficulty quantification. By focusing attention on semantically rich yet challenging regions, HSM establishes a label-free paradigm for hardness assessment in self-supervised learning.

- We propose Hard Region Discovery (HRD), a reinforcement learning-based module that employs policy gradient optimization to adaptively select and reconstruct critical intra-sample regions. Unlike conventional random masking, HRD strategically prioritizes challenging yet discriminative areas, forcing the model to focus on semantically rich facial dynamics.

## 2 RELATED WORK

### 2.1 DYNAMIC FACIAL EXPRESSION RECOGNITION

The rapid advancement of deep learning has significantly propelled research in DFER. Existing approaches can be broadly categorized into three paradigms. First, 2D CNN-based methods Fan et al. (2016); Liu et al. (2018); Wang et al. (2020) extract spatial features from individual frames and subsequently model temporal dependencies with RNNs. Second, 3D CNN approaches Wang et al. (2023); Jiang et al. (2020); Liu et al. (2021); Wang et al. (2022a) directly learn spatiotemporal representations from videos. Recently, Transformer-based architectures Li et al. (2023); Wang et al. (2022b); Zhao & Liu (2021); Ma et al. (2022) jointly model spatial and temporal dynamics. However, these methods rely on supervised learning paradigms, which remain constrained by the limited scale of annotated samples in current DFER datasets.

To address this challenge, several CLIP variants Zhao & Patras (2023); Foteinopoulou & Patras (2024); Li et al. (2024a); Mai et al. (2024); Li et al. (2024b) have demonstrated promising transferability and generalization capabilities. However, these approaches lack explicit spatiotemporal modeling mechanisms, limiting their effectiveness in capturing dynamic facial expression variations. Recent self-supervised learning methods have attempted to leverage large-scale unlabeled video data. MAE-DFER Sun et al. (2023) leverages masked reconstruction to simultaneously learn facial appearance features and temporal dynamics. HiCMAE Sun et al. (2024) combines masked modeling with contrastive learning objectives to enhance feature representations at multiple semantic levels. Further extending this direction, AVF-MAE++ Wu et al. (2025) develops an adaptive dual-masking mechanism that operates across audio-visual modalities, establishing an efficient cross-modal self-supervised learning paradigm. A key limitation of these methods is their uniform treatment of all facial regions and samples, neglecting the inherent variation in their importance for DFER. This uniform attention approach may lead to suboptimal learning of discriminative spatiotemporal features.

### 2.2 MASKED AUTOENCODERS

Recent advances in masked autoencoders have demonstrated critical impacts of masking strategies on downstream task performance Bao et al. (2022); He et al. (2022); Tong et al. (2022). Various random masking approaches have been explored, including grid, block, and patch masking for images Bao et al. (2022); Zhou et al. (2022), as well as pipeline, frame, and patch masking for videos Tong et al. (2022); Shi et al. (2022); Li et al. (2022). However, these static or stochastic masking strategies often overlook crucial dynamic regions and subtle expression variations that contain discriminative information.

The limitations of current approaches become evident when examining their inconsistent performance across different datasets. For example, VideoMAE Tong et al. (2022) achieves optimal action classification results on SSv2 Goyal et al. (2017) using pipeline masking, while Spatio-Temporal MAE Shi et al. (2022) performs best on Kinetics-400 Kay et al. (2017) with patch masking. These performance variations likely stem from fundamental differences in dataset characteristics, collection methodologies, and the inherent distribution of spatiotemporal features. These observations highlight the need for sophisticated masking strategies that can automatically identify and prioritize challenging regions and samples. An adaptive selection mechanism that focuses on complex facial dynamics and subtle expression changes could significantly improve model robustness and generalization abilities across diverse datasets and real-world conditions.

## 3 METHODS

Figure 2 presents the architecture of our proposed HI2M framework, comprising two core modules: (1) Hard Sample Mining (HSM), which identifies challenging samples to enhance reconstruction learning, and (2) Hard Region Discovery (HRD), which automatically locates discriminative facial regions for adaptive masking.

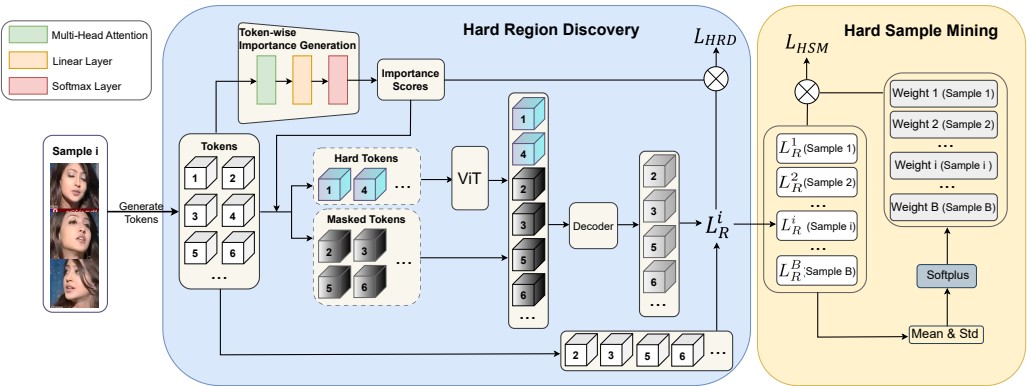

Figure 2: Overview of our proposed HI2M model which mainly consists of Hard Sample Mining (HSM) and Hard Region Discovery (HRD). The HSM dynamically adjusts the importance scores of different samples based on their reconstruction loss values. It then targets at reconstructing more challenging samples. The HRD is an adaptive and trainable token selection mechanism that enhances the identification of informative spatiotemporal regions within the frame by leveraging intra-sample similarity scores.

## 3.1 PRELIMINARIES

For video processing, we first partition an input video of dimensions $T \times 3 \times H \times W$ into non-overlapping cubic tokens of size $2 \times 3 \times 16 \times 16$, where $T$ represents the number of frames, and $H$ and $W$ denote the height and width of each frame. Each frame contains RGB channels. A 3D convolutional layer with kernel size $(2, 3, 16, 16)$, stride $(2, 16, 16)$, and $d$ output channels is employed to generate tokens $X$. This results in $N = \frac{T}{2} \times \frac{H}{16} \times \frac{W}{16}$ tokens. Each token is with dimensionality $d$ (set to 384 in our design). Positional information is integrated using a fixed 3D periodic positional encoding scheme Vaswani et al. (2017), enhancing the model's capacity to capture both spatial and temporal dependencies.

## 3.2 HARD SAMPLE MINING

Effective prioritization of challenging samples is crucial for enhancing model robustness in complex video understanding tasks. To address this, we propose a Hard Sample Mining (HSM) mechanism that dynamically weights training samples according to their reconstruction loss distribution.

To jointly model both appearance and motion characteristics during reconstruction, we define a masked reconstruction loss $L_R$, which is based on the mean squared error between the reconstructed tokens and their corresponding ground-truth RGB values after patch normalization.

During pre-training, we use a dual-target reconstruction strategy, where the decoder's output tokens are divided into two streams—one for appearance targets and the other for motion targets. The total reconstruction loss computes squared differences between predicted tokens $\hat{V}_i$ and both appearance $V_i^a$ and motion $V_i^m$ targets, sums these squared errors, and normalizes by the number of selected tokens $N - N_v$, as follows:

$$L_R = \frac{1}{N - N_v} \sum_{i \in I_m} \left( \left| \hat{V}_i - V_i^a \right|^2 + \left| \hat{V}_i - V_i^m \right|^2 \right), \tag{1}$$

where $I_m$ represents the set of selected token indices, and $\hat{V}_i$ is the predicted token at index $i$. The appearance target $V_i^a$ is the odd frames from the video sequence $V[0 : T : 2]$. The motion target $V_i^m$ is computed as the difference between adjacent frames, by subtracting odd frames from even frames $V[1 : T : 2] - V[0 : T : 2]$. This approach enables the simultaneous learning of both static and dynamic representations within a single decoder structure, where appearance and motion features are reconstructed separately.

Specifically, the mean $\mu$ and standard deviation $\sigma$ of the reconstruction loss is firstly calculated across the batch, as follows:

$$\mu = \frac{1}{B}\sum_{i=1}^{B} L_R^i, \quad \sigma = \sqrt{\frac{1}{B}\sum_{i=1}^{B}(L_R^i - \mu)^2}, \tag{2}$$

where $B$ denotes the number of images in a batch and $L_R^i$ is the reconstruction loss of the $i$-th sample.

Then, the weight $w_i$ is determined by the deviation of $L_R^i$ from the batch-wise loss distribution. In order to ensure a stable and robust training process, the smooth nature of the Softplus function is used to avoid abrupt optimization dynamics. The formula is defined as follows:

$$w_i = \log\left(1 + e^{\frac{L_R^i - \mu}{\sigma}}\right). \tag{3}$$

Finally, the loss $L_{HSM}$ is defined as follows:

$$L_{HSM} = \sum_i w_i L_R^i. \tag{4}$$

Our HSM automatically identifies and emphasizes difficult cases—including occluded regions and frames with rapid temporal changes—enabling the model to focus its learning capacity on the most informative video segments. Unlike conventional approaches using fixed thresholds or manual hyperparameter tuning, our HSM automatically adapts sample weights based on batch statistics.

### 3.3 HARD REGION DISCOVERY

Conventional masking strategies treat uniformly all regions regardless of discriminative value. To address this, we propose Hard Region Discovery (HRD), an adaptive masking mechanism that automatically identifies and prioritizes complex and high-information regions.

Firstly, to dynamically determine masking candidates, we employ a token-wise importance generation network, parameterized by $\theta$, which consist of multi-head attentions on tokens $X$ coupled with linear and Softmax layers, yielding token-wise importance scores $P \in \mathbb{R}^N$.

Secondly, we formulate token selection as a reinforcement learning (RL) problem and adopt policy gradient methods to optimize the masking strategy. In RL, the agent iteratively observes a state, selects an action, and receives a reward to learn an optimal policy that maximizes cumulative returns. We frame HRD as an RL problem: the token-wise selection network acts as the agent, whose action is deciding whether to select a token, while the reward is defined as the reconstruction loss. By maximizing the expected reward, RL enables end-to-end training of HRD, prioritizing semantically challenging regions (e.g., high-information tokens). The policy gradient update rule optimizes the selection network parameters $\theta$ via:

$$\Delta\theta = \alpha R \frac{\partial \log P(t)}{\partial \theta}, \tag{5}$$

where the reward $R$ is defined as the reconstruction error, and $P(t)$ denotes the selection probability of token $t$, modeled by a categorical distribution $P$. The learning rate $\alpha$ determines the step size of the policy updates. The objective maximizes the log-likelihood of selected tokens under reward guidance, refining the policy to focus on hard regions.

The categorical distribution $P$ serves as a dynamic sampling policy, prioritizing more informative tokens. We sample visible token indices from $P$ based on their reconstruction difficulty:

$$I_v \sim \text{Categorical}(N_v, P), \tag{6}$$

where $I_v$ denotes selected visible tokens. The number of visible tokens is determined by the masking ratio $\rho$ with $N_v = N \times (1 - \rho)$.

Thirdly, to optimize computational efficiency, only the visible (non-masked) tokens are fed into the encoder. We adopt the standard ViT-Base architecture with space-time attention Bertasius et al. (2021), allowing the attention mechanism to focus on the visible tokens and capture contextual information, ensuring that only the most informative tokens influence the final output. The encoded

visible tokens, together with masked tokens, are forwarded to the decoder. The masked patches are treated as learnable tokens and the decoder reconstructs them by minimizing the MSE loss between the predicted and ground-truth values. The decoder reconstructs the original video cube of size $\frac{T}{2} \times \frac{H}{16} \times \frac{W}{16}$ from both encoded visible and masked tokens.

Finally, to optimize the hard region discovery process, a loss $L_{HRD}$ is introduced to increase the probability of selecting difficult tokens:

$$L_{HRD} = -\sum_j \sum_{i \in I_m^j} log P_i^j \cdot L_R^{i,j}, \tag{7}$$

where $P_i^j$ is the probability of masking token $i$ for sample $j$, $L_R^{i,j}$ represents the reconstruction loss for token $i$ in sample $j$, and $I_m^j$ is the masked $j$-th sample. The negative sign ensures proper gradient optimization. To stabilize training and avoid numerical instability due to small $P_i$ values, we apply a logarithmic transformation to $P$. Additionally, we ensure gradients do not backpropagate through $L_R$ into the MAE model, enabling independent learning of the hard region discovers.

### 3.4 JOINT TRAINING OBJECTIVES

Effective model training requires careful balance between HRD and HSM, as independent optimization of these mechanisms yields suboptimal results. Exclusive focus on token-level difficulty risks incorporating noise from easily reconstructible samples, while pure sample reweighting may neglect critical spatial details. To address this, we introduce a unified objective function that jointly optimizes:

$$L = L_{HSM} + \lambda L_{HRD}, \tag{8}$$

where $\lambda$ controls the balance between the HSM loss $L_{HSM}$ and HRD loss $L_{HRD}$.

Unlike standard MAE training that applies uniform masking, our framework introduces two complementary difficulty-aware mechanisms: (1) sample-level weighting based on reconstruction difficulty, and (2) region-level adaptive masking guided by semantic importance. This dual-level approach ensures balanced attention to both challenging samples and discriminative spatial features during representation learning.

## 4 EXPERIMENTS

### 4.1 DATASETS AND MEASUREMENTS

**Pre-training Dataset.** We conduct pre-training using the VoxCeleb2 dataset Chung et al. (2018), which contains over 1 million video clips featuring more than 6,000 speakers extracted from approximately 150,000 YouTube interview videos. Following standard protocol, we use only the development set (1,092,009 clips from 145,569 videos) for pre-training, reserving the test set for evaluation.

**DFER Datasets.** Two benchmark datasets are used: DFEW Jiang et al. (2020) (12,059 movie clips with challenging conditions including lighting variations, occlusions, and diverse poses, annotated by 10 evaluators for seven emotions using 5-fold cross-validation) and FERV39K Wang et al. (2022a) (the largest DFER dataset with 38,935 clips across 22 fine-grained scenes, professionally annotated by 30 evaluators per clip for the same seven emotions, using official train-test splits).

**Performance Measurements.** Following previous studies, we report both unweighted average recall (UAR) and weighted average recall (WAR).

### 4.2 IMPLEMENTATION DETAILS

**Pre-training.** The original resolution of videos in the VoxCeleb2 dataset is $224 \times 224$. For each frame, a $160 \times 160$ crop is extracted from the upper-center region to consistently focus on the speaker's facial area. During pre-training, 16 frames are uniformly sampled from each video clip with a temporal stride of 4, balancing motion continuity and computational efficiency. We employ the AdamW optimizer with momentum parameters ($\beta_1 = 0.9, \beta_2 = 0.95$), using a base learning rate of $3 \times 10^{-4}$ and weight decay of 0.05. Training proceeds for 80 epochs with a 5-epoch linear warmup

period, followed by cosine learning rate decay. Experiments are conducted with a batch size of 60 across two NVIDIA RTX 3090 GPUs.

**Fine-tuning.** During fine-tuning, the input size remains $16 \times 160 \times 160$ (frames × height × width), with dataset-specific temporal strides: 4 for DFEW to capture broader temporal context and 1 for FERV39K to preserve fine-grained motion. Fine-tuning also uses the AdamW optimizer ($\beta_1 = 0.9, \beta_2 = 0.999$) with an initial learning rate of $1 \times 10^{-3}$ and a batch size of 24. The training runs for 100 epochs, including a 5-epoch linear warm-up. We fine-tune the pre-trained VideoMAE model for DFER by adding a linear classifier on top of its frozen features. This classifier is trained to map the general representations to the specific expression categories.

## 4.3 COMPARISON WITH THE STATE-OF-THE-ART METHODS

As shown in Table 1, we compare our model against SOTA approaches on both DFEW and FERV39K, reporting results as (WAR, UAR) pairs for conciseness. While existing methods—including 3D CNN models Wang et al. (2023) and Transformer frameworks Zhao & Liu (2021); Wang et al. (2022b); Li et al. (2023)—have advanced DFER performance, they are purely supervised learning paradigms.

Table 1: Performance comparison between our HI2M and state-of-the-art methods. **Bold** values indicate the best results; Underlined values denote the second-best results.

| Methods | Venue | DFEW | | FERV39k | |
| --- | --- | --- | --- | --- | --- |
| | | WAR | UAR | WAR | UAR |
| Former-DFERZhao & Liu (2021) | MM'21 | 65.70 | 53.69 | 46.85 | 37.20 |
| DPCNetWang et al. (2022b) | MM'22 | 66.32 | 57.11 | - | - |
| IALLi et al. (2023) | AAAI'23 | 69.24 | 55.71 | 48.54 | 35.82 |
| M3DFELWang et al. (2023) | CVPR'23 | 69.25 | 56.10 | 47.67 | 35.94 |
| DFER-CLIPZhao & Patras (2023) | BMVC'23 | 71.25 | 59.61 | 51.65 | 41.27 |
| MAE-DFERSun et al. (2023) | MM'23 | 74.43 | 63.41 | 52.07 | 43.12 |
| EmoCLIPFoteinopoulou & Patras (2024) | FG'24 | 62.12 | 58.04 | 36.18 | 31.41 |
| CLIPERLi et al. (2024a) | ICME'24 | 70.84 | 57.76 | 51.34 | 41.23 |
| HiCMAESun et al. (2024) | IF'24 | 73.10 | 61.92 | - | - |
| UMBEnetMai et al. (2024) | MM'24 | 73.93 | 64.55 | 52.10 | 44.01 |
| DK-CLIPLi et al. (2024b) | MM'24 | 75.41 | 64.95 | 52.14 | 43.71 |
| IFDD-3DViTWang & Chai (2025) | AAAI'25 | 73.82 | 61.19 | 51.09 | 39.15 |
| HDF Cui et al. (2025) | MM'25 | 71.60 | 60.40 | 50.30 | 40.49 |
| AVF-MAE++Wu et al. (2025) | CVPR'25 | 75.42 | 63.74 | - | - |
| HI2M | Ours | **75.85** | **66.42** | **52.92** | **44.40** |

To address these, utilizing large-scale unlabeled data offers a viable solution. Existing MAE approaches Sun et al. (2023; 2024); Wu et al. (2025) treat all regions and samples equally, resulting in sub-optimal representations due to incorporated noise. However, our HI2M model achieves superior performance through adaptive selection of informative spatiotemporal features. On DFEW, HI2M attains (75.85%, 66.42%) in (WAR, UAR), outperforming AVF-MAE Wu et al. (2025) (75.42%, 63.74%) by +0.43% WAR and +2.68% UAR. Similarly, on FERV39K, our method reaches (52.92%, 44.40%), surpassing MAE-DFER Sun et al. (2023) (52.07%, 43.12%) by +0.85% WAR and +1.28% UAR. Unlike previous methods, HI2M's targeted reconstruction of discriminative regions enables more effective learning of cross-spatiotemporal relationships essential for DFER.

Recent variants of vision-language models Zhao & Patras (2023); Foteinopoulou & Patras (2024); Li et al. (2024a); Mai et al. (2024); Li et al. (2024b) based on CLIP Radford et al. (2021) have demonstrated strong generalization in DFER despite being pre-trained without facial videos. However, their lack of explicit spatiotemporal modeling limits their ability to capture dynamic expression changes across frames. This explains the superior performance of our HI2M model, which achieves 75.85% WAR and 66.42% UAR on DFEW, outperforming the best CLIP-based method DK-CLIP Li et al. (2024b) (75.41%, 64.95%) by +0.44% WAR and +1.47% UAR. On FERV39K, HI2M (52.92%, 44.40%) surpasses DK-CLIP (52.14%, 43.71%) by +0.78% WAR and +0.69% UAR. Our results show that the proposed approach not only benefits from large-scale masked modeling but also excels at capturing fine-grained spatiotemporal dependencies, which are crucial for achieving robust DFER.

## 4.4 ABLATION STUDY

We perform component analysis using 5-fold cross-validation on the DFEW dataset. We also evaluate the effectiveness of pre-training by applying HI2M under three training strategies—random

initialization, linear probing, and full fine-tuning—on both DFEW and FERV39K. This experimental design enables a comprehensive analysis of model behavior across different dimensions.

**Effects of Different Components.** As shown in Table 2, we conduct a ablation study to evaluate the contributions of our core modules—Hard Sample Mining (HSM) and Hard Region Discovery (HRD)—by progressively incorporating each component. The baseline model (without HRD/HSM components) achieves (74.67%, 63.35%) in (WAR, UAR), demonstrating the fundamental limitations of uniform treatment: (1) inability to prioritize challenging facial regions containing subtle expression cues, and (2) suboptimal handling of class-imbalanced samples. This performance gap highlights the necessity of adaptive mechanisms for capturing discriminative spatiotemporal features in DFER.

The integration of HRD yields measurable improvements, increasing performance to (75.16%, 64.59%) in (WAR, UAR). This 1.24% UAR gain demonstrates that targeted reconstruction of challenging facial regions—particularly those containing subtle expression cues like micro-movements around the eyes and mouth—enhances feature discriminability. However, HRD's region-level optimization alone cannot address sample-wide difficulty variations, as it maintains uniform weighting across all training instances, leaving potential gains from hard sample mining unexploited.

Table 2: Evaluations about different components.

| Modules | | Metrics(%) | |
|---|---|---|---|
| HSM | HRD | WAR | UAR |
| - | - | 74.67 | 63.35 |
| - | ✓ | 75.16 | 64.59 |
| ✓ | - | 75.46 | 65.15 |
| ✓ | ✓ | **75.85** | **66.42** |

The HSM, which adaptively weights samples according to reconstruction difficulty, yields significant performance gains of (+0.79% WAR, +1.80% UAR) over the baseline, achieving (75.46%, 65.15%). The substantially larger UAR improvement (1.80% vs 0.79% WAR) further indicates that sample weighting particularly benefits minority class recognition without compromising overall accuracy.

Overall, our hierarchical integration of HRD and HSM demonstrates their complementary advantages for DFER. HRD operates at the local level through adaptive region masking, selectively reconstructing discriminative facial dynamics (e.g., periorbital and perioral muscle movements), while HSM addresses global dataset bias via difficulty-aware sample weighting. The combined framework yields synergistic performance gains of (+0.69% WAR, +1.83% UAR) over HRD alone and (+0.39% WAR, +1.27% UAR) over HSM alone, with three key implications: (1) DFER benefits from dual-level (local/global) difficulty adaptation, (2) region selection and sample reweighting constitute orthogonal improvement axes, and (3) hierarchical hard example mining represents an effective paradigm for learning robust facial representations from limited labeled data.

Table 3: Performance comparison of different training strategies on DFEW and FERV39K datasets.

| Dataset | Metric | Random Init. | Linear Probing | Full Fine-tuning |
|---|---|---|---|---|
| DFEW | WAR | 22.85 | 51.98 | 75.85 |
| | UAR | 14.31 | 38.13 | 66.42 |
| FERV39K | WAR | 24.59 | 42.72 | 52.92 |
| | UAR | 14.58 | 29.73 | 44.40 |

**Evaluation of Pre-training Effectiveness across Training Strategies.** To assess the effectiveness of the representations learned during pre-training, we perform linear probing by freezing the encoder and training only a linear classifier. Results are reported in Table 3. Compared to random initialization, linear probing yields substantial improvements, demonstrating the effectiveness of our reconstruction-based pre-training in learning transferable and discriminative features. While full fine-tuning achieves the best performance, the gap between linear probing and fine-tuning is more pronounced on DFEW, likely due to its limited diversity and scale. In contrast, FERV39K's larger and more varied samples enable linear classifiers to perform competitively, even without updating the encoder.

## 4.5 VISUALIZATION ANALYSIS

**Confusion Matrix.** To comprehensively evaluate our method's effectiveness, we conduct visualization analyses on FERV39K and DFEW (Fold 1) datasets. This dual-dataset assessment serves two key purposes: (1) verifying consistent attention to discriminative facial regions across different data distributions, and (2) examining the model's robustness to varying recording conditions and expression intensities. Analysis of the confusion matrix in Figure 3 reveals several key insights: (1) Performance varies significantly by emotion category, with "Happiness" showing the highest accuracy

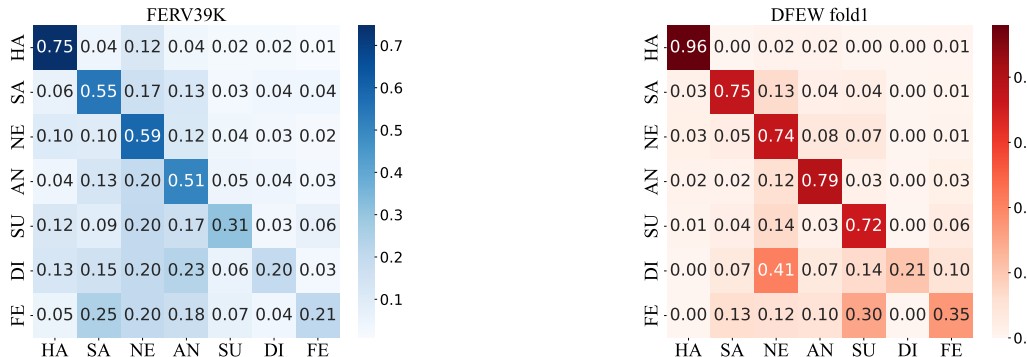

Figure 3: Confusion matrix for our HI2M model, with predicted classes on the horizontal axis and ground-truth labels on the vertical axis.

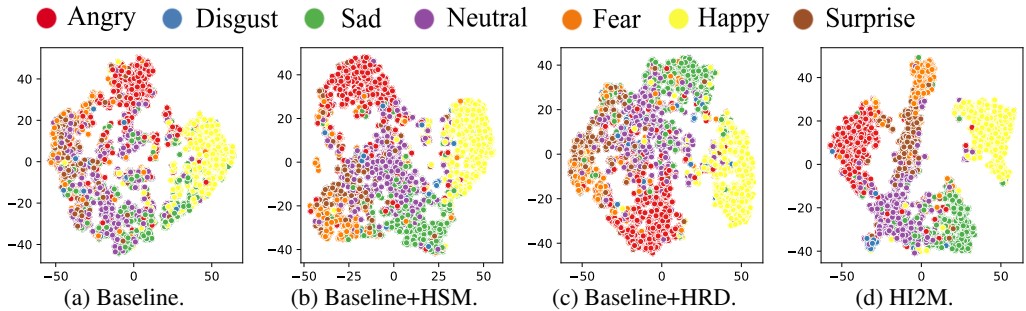

(a) Baseline.     (b) Baseline+HSM.     (c) Baseline+HRD.     (d) HI2M.

Figure 4: t-SNE visualization of feature distributions for (a) Baseline, (b) Baseline+HSM, (c) Baseline+HRD, and (d) HI2M.

while "Disgust" suffers from poor recall due to label imbalance; (2) A pronounced bias toward "Neutral" predictions suggests the model defaults to this category for low-intensity expressions, indicating an opportunity to improve subtle emotion detection through better intensity-sensitive features.

**t-SNE Visualization.** We apply t-SNE van der Maaten & Hinton (2008) to visualize the feature distribution of the baseline models, baseline+HSM, baseline+HRD, and our proposed HI2M. As shown in Figure 4, the features of the baseline model overlap significantly across categories, indicating limited discriminative power. In contrast, HI2M produces clearer boundaries between categories, with more compact and distinct clusters. However, some overlap remains, especially for "Neutral" expressions. This overlap is a well-known challenge in DFER, as low-intensity expressions are difficult to distinguish and are often confused with the "Neutral" category.

## 5 CONCLUSIONS

We present Hard Intra- and Inter-sample Masked Modeling (HI2M), a novel masked autoencoder framework for dynamic facial expression recognition (DFER). HI2M introduces two key innovations: (1) Hard Sample Mining (HSM) that dynamically weights samples based on reconstruction difficulty to prioritize challenging cases, and (2) Hard Region Discovery (HRD) that employs learnable attention to identify and emphasize discriminative spatiotemporal regions. Unlike conventional approaches using random or fixed masking patterns, HI2M's hierarchical adaptation - operating simultaneously at both sample and region levels - enables more effective modeling of subtle facial dynamics by establishing robust relationships between critical expression features. Comprehensive experiments on benchmark datasets demonstrate HI2M's superiority, achieving state-of-the-art performance while providing interpretable attention patterns that align with known facial action units.

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

## A    THE USE OF LLMs

Yes, the authors utilized a Large Language Model exclusively to enhance the academic writing quality. Specifically, the tool was applied to refine grammatical accuracy, improve sentence fluency, and strengthen overall textual clarity after the core intellectual content had been fully developed by the authors.

All scientific contributions - including the conceptualization of the proposed methodology, experimental design, result analysis, and conclusion derivation - remain exclusively attributable to the human authors. The model was not involved in any aspects related to data generation, algorithmic development, or technical reasoning. All content underwent thorough review and editing by the authors to ensure accurate representation of the research conducted.

## B    EFFECTS OF LOSS BALANCE COEFFICIENT $\lambda$

Table 4: Effects of the loss balance coefficient $\lambda$ on model performance.

| Dataset | Metrics (%) | $\lambda$ | | | | |
|---|---|---|---|---|---|---|
| | | $10^{-5}$ | $5 \times 10^{-4}$ | $10^{-4}$ | $5 \times 10^{-3}$ | $10^{-3}$ |
| DFEW fold1 | WAR | 74.48 | 75.31 | **75.74** | 74.88 | 74.75 |
| | UAR | 63.63 | 63.61 | **64.49** | 62.84 | 62.65 |
| FERV39k | WAR | 50.13 | 51.31 | **52.92** | 51.73 | 50.29 |
| | UAR | 41.40 | 41.66 | **44.40** | 41.86 | 41.49 |

Table 4 summarizes the ablation study evaluating the effect of different $\lambda$ values in Eqn. 8 on both the FERV39k and DFEW datasets. Compared to random region masking (i.e., $\lambda = 0$), introducing the HRD loss ($L_{HRD}$) helps the model focus on semantically challenging regions, thereby enhancing feature discriminability. However, a large $\lambda$ (i.e., $10^{-3}$) induces training instability, while a too-small $\lambda$ (i.e., $10^{-5}$) yields only marginal improvements. The optimal trade-off is consistently observed at $\lambda = 10^{-4}$, which achieves the highest WAR and UAR on both datasets. This value effectively balances the complementary mechanisms of HSM (sample-level weighting) and HRD (region-level adaptive masking), ensuring that the model prioritizes both challenging samples and discriminative spatial features—aligning well with the design of our joint training objective. The generalizability of $\lambda = 10^{-4}$ across DFEW fold1 and FERV39k confirms its robustness.

## C    EFFECT OF PRE-TRAINING EPOCHS ON MODEL PERFORMANCE

Table 5: Ablation study on the number of pre-training epochs.

| Dataset | Metric | 0 | 25 | 50 | 60 | 70 | 80 |
|---|---|---|---|---|---|---|---|
| DFEW | WAR | 22.85 | 73.52 | 74.14 | 75.31 | **75.74** | 75.01 |
| | UAR | 14.31 | 60.27 | 62.8 | 62.65 | **64.49** | 63.16 |
| FERV39K | WAR | 24.59 | 52.5 | 52.84 | 52.92 | **53.17** | 52.8 |
| | UAR | 14.58 | 42.5 | 42.88 | **44.40** | 43.14 | 43.4 |

Table 5 demonstrates the impact of pre-training epochs on model performance. The results indicate dataset-dependent optimal epochs: DFEW fold 1 performs best at 70 epochs (75.74% WAR, 64.49% UAR), while FERV39K achieves peak UAR (44.40%) at 60 epochs and highest WAR (53.17%) at 70 epochs. Crucially, skipping pre-training (0 epochs) reduces performance to random-guessing levels, demonstrating that self-supervised pre-training is indispensable when labeled data is scarce.

Table 6: Per-category performance comparison on DFEW between our method and state-of-the-art approaches. Standard deviation (Std) of per-class accuracy is reported. **Bold** and Underlined values indicate the best and second-best results, respectively.

| Methods | Metrics(%) | | Accuracy of Each Emotion (%) | | | | | | | Std |
|---|---|---|---|---|---|---|---|---|---|---|
| | WAR | UAR | Happy | Sad | Neutral | Anger | Surprise | Disgust | Fear | |
| Former-DFERZhao & Liu (2021) | 65.70 | 53.69 | 84.05 | 62.57 | 67.52 | 70.03 | 56.43 | 3.45 | 31.78 | 25.32 |
| IALLi et al. (2023) | 69.24 | 55.71 | 87.95 | 67.21 | 70.10 | 76.06 | 62.22 | 0.00 | 26.44 | 28.78 |
| M3DFELWang et al. (2023) | 69.25 | 56.10 | 89.59 | 68.38 | 67.88 | 74.24 | 59.69 | 0.00 | 31.63 | 28.03 |
| MARLINCai et al. (2023) | 66.74 | 53.94 | 85.77 | 66.64 | 67.22 | 69.54 | 60.72 | 0.00 | 27.72 | 24.11 |
| DFER-CLIPZhao & Patras (2023) | 71.25 | 59.61 | 91.12 | 75.34 | 71.15 | 74.09 | 56.30 | 11.72 | 37.81 | 24.96 |
| MAE-DFERSun et al. (2023) | 74.43 | 63.41 | 92.92 | **77.46** | 74.56 | 76.94 | 60.99 | 18.62 | 42.35 | 23.39 |
| DK-CLIPLi et al. (2024b) | 75.41 | 64.95 | **94.61** | 76.19 | **75.06** | 78.65 | 63.61 | 23.60 | 42.93 | 22.33 |
| HI2M (Ours) | **75.85** | **66.42** | 93.78 | 77.14 | 72.18 | **78.92** | **69.70** | **28.17** | **43.41** | **20.89** |

# D  QUANTITATIVE RESULTS ACROSS EMOTION CATEGORIES

Our evaluation reveals key insights into emotion-specific performance on the DFEW dataset (Table C). The proposed method demonstrates state-of-the-art results across most categories, with particularly notable gains for challenging expressions: "Surprise" (+6.09% absolute improvement) and "Disgust" (+4.57%). These improvements are especially significant given (1) the inherent complexity of their facial dynamics, and (2) their underrepresentation in training data ("Disgust": 1.22%, "Surprise": 12.42%; Table 8a). The model's ability to excel on these sparse yet complex categories highlights its data-efficient learning capability. Furthermore, our approach achieves the lowest standard deviation (20.89) among all compared methods, demonstrating superior consistency across emotion categories.

The results on FERV39K (Table D) further demonstrate our method's robust generalization under a more challenging setting. Compared to prior approaches, HI2M achieves the highest WAR and UAR, along with top performance on most categories, including "Happy" (+1.70%), "Sad" (+0.58%), "Anger" (+0.47%), "Surprise" (+0.94%), "Disgust" (+1.71%), and "Fear" (+3.71%). Again, as indicated in Table 8b, these categories—especially "Disgust" (5.91%) and "Fear" (5.42%)—are less frequent than others like "Neutral" or "Angry", yet our method maintains strong recognition ability on them. While performance on "Neutral" is marginally lower than MAE-DFER, our approach yields the smallest standard deviation (19.35), suggesting a more balanced and stable recognition ability across both frequent and underrepresented emotions.

Table 7: Per-category performance comparison on FERV39k between our method and state-of-the-art approaches. Standard deviation (Std) of per-class accuracy is reported. **Bold** and Underlined values indicate the best and second-best results, respectively.

| Methods | Metrics(%) | | Accuracy of Each Emotion (%) | | | | | | | Std |
|---|---|---|---|---|---|---|---|---|---|---|
| | WAR | UAR | Happy | Sad | Neutral | Anger | Surprise | Disgust | Fear | |
| Former-DFERZhao & Liu (2021) | 46.85 | 37.20 | 65.65 | 51.33 | 56.74 | 43.64 | 21.94 | 8.57 | 12.53 | 21.02 |
| MAE-DFERSun et al. (2023) | 52.07 | 43.12 | 73.05 | 53.98 | **59.14** | 50.44 | 30.09 | 17.99 | 17.17 | 19.98 |
| HI2M (Ours) | **52.92** | **44.40** | **74.75** | **54.56** | 58.99 | **50.91** | **31.03** | **19.70** | **20.88** | **19.35** |

Our method achieves robust performance not only in recognizing high-intensity emotions (e.g., "Anger", "Surprise") but also in detecting subtle expressions (e.g., "Neutral", "Sad"), demonstrating balanced modeling of both distinct and ambiguous categories. The consistently lower standard deviation across datasets further indicates stable performance regardless of category frequency or complexity. These improvements stem from the combined contribution of two components: HRD and HSM. HRD dynamically focuses on semantically informative facial regions through a learnable masking policy, optimized via reinforcement learning and spatiotemporal constraints. Meanwhile, HSM identifies and prioritizes challenging samples, mitigating class imbalance and enhancing learning on rare or complex expressions. Together, these mechanisms enable HI2M to adaptively capture discriminative features, improving accuracy and generalization across diverse facial expressions.

Table 8: Emotion category distribution in (a) DFEW and (b) FERV39K datasets.

| Emotion | Number of Clips | Percentage (%) | Emotion | Number of Clips | Percentage (%) |
|---------|-----------------|----------------|---------|-----------------|----------------|
| Happy | 2488 | 20.63 | Happy | 7327 | 18.82 |
| Sad | 2008 | 16.65 | Sad | 6916 | 17.76 |
| Neutral | 2709 | 22.46 | Neutral | 9748 | 25.03 |
| Angry | 2229 | 18.48 | Angry | 7393 | 18.98 |
| Surprise | 1498 | 12.42 | Surprise | 3140 | 8.06 |
| Disgust | 146 | 1.22 | Disgust | 2300 | 5.91 |
| Fear | 981 | 8.14 | Fear | 2111 | 5.42 |
| Total | 12059 | 100.00 | Total | 38935 | 100.00 |
| (a) DFEW Dataset | | | (b) FERV39K Dataset | | |

## E   EFFECTS OF DIFFERENT MASKING RATIO ($\rho$) VALUES ON MODEL PERFORMANCE

Figure 5 reveals how varying masking ratios affect (WAR/UAR) performance on DFEW and FERV39K datasets. Three key observations emerge: (1) DFEW fold 1 achieves peak performance at a 0.8 masking ratio (75.74% WAR, 64.49% UAR), suggesting this ratio optimally balances information preservation and learning challenge; (2) FERV39K shows different characteristics, with WAR peaking at 53.13% for 0.95 masking ratio while UAR reaches maximum (44.4%) at 0.9; (3) Both datasets demonstrate performance degradation at suboptimal ratios, confirming the importance of ratio selection. This dataset-dependent behavior highlights how architectural properties and data complexity influence the ideal masking strategy.

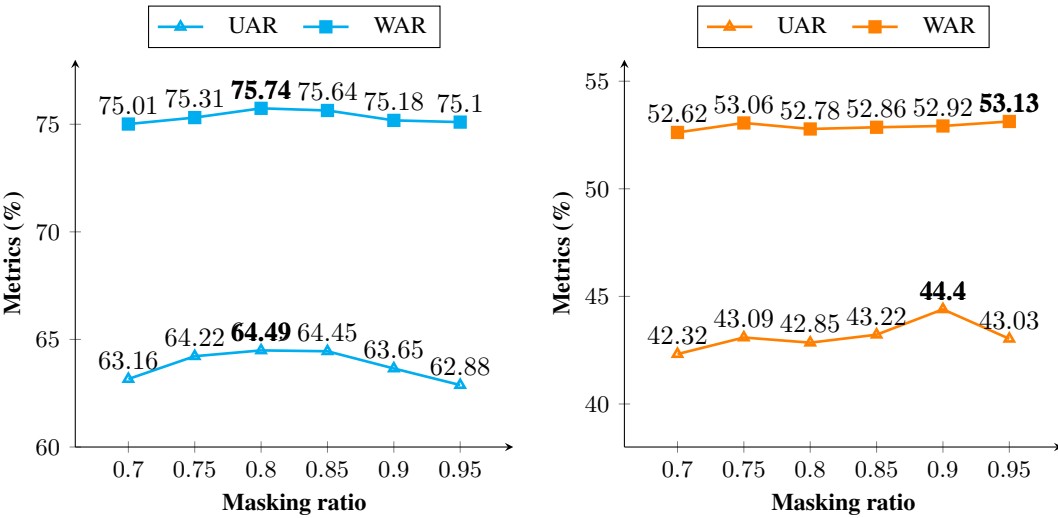

Figure 5: Performance comparison under different masking ratios ($\rho$) on DFEW fold 1 and FERV39K.

## F   CROSS-FOLD PERFORMANCE ANALYSIS

To thoroughly validate the model's generalization capability, we analyze its performance across fold 2 through fold 5 of the DFEW dataset, as shown in Figure 6. The confusion matrices and evaluation metrics reveal several key aspects of the model's behavior across different data partitions.

The results demonstrate consistent performance across all folds, with WAR ranging from 74.03% to 77.23% and UAR between 63.87% and 70.62%. This stability in both overall and per-class metrics indicates robust generalization beyond specific data splits.

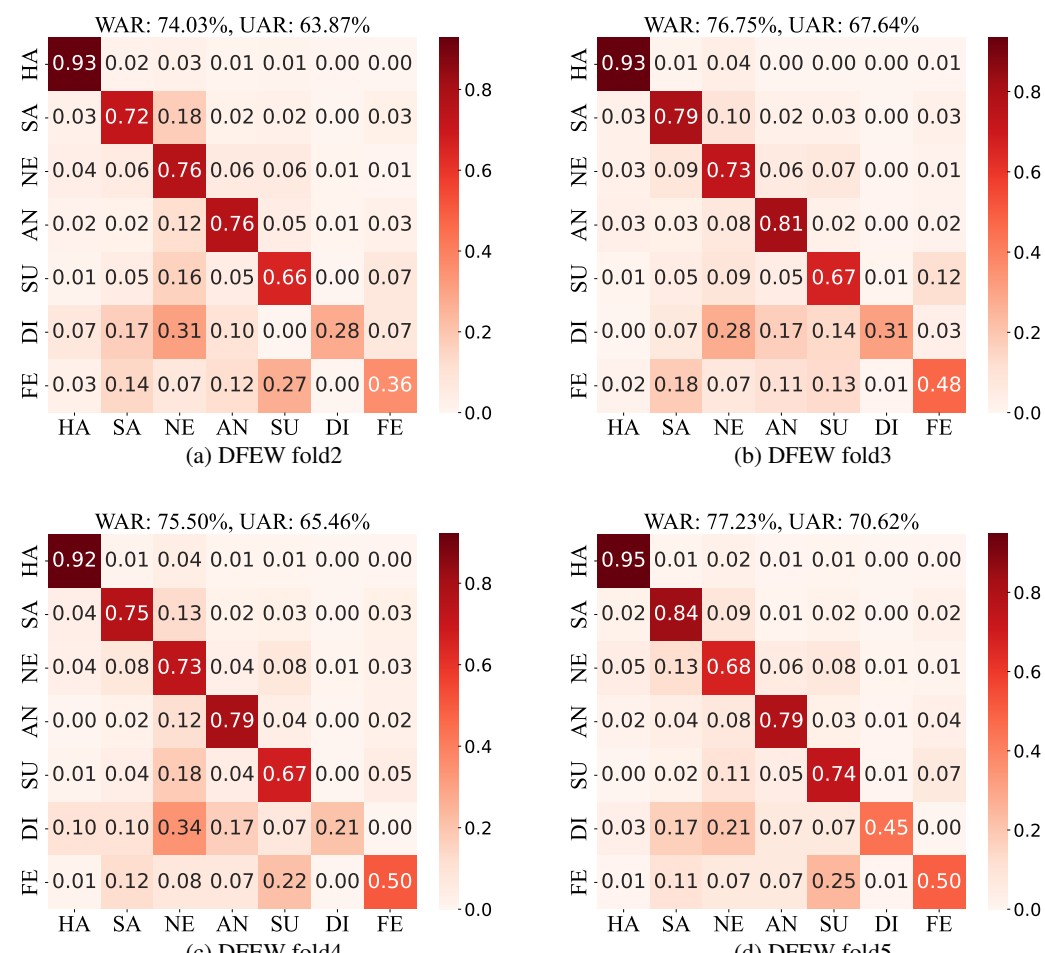

Figure 6: Confusion matrices for our model on DFEW folds 2-5. Ground truth labels (vertical) versus predictions (horizontal).

More importantly, we observe substantial improvements in traditionally challenging categories. The "Disgust" recognition reaches 45% accuracy in fold 5. Similarly, "Fear" classification achieves 50% accuracy in folds 4 and 5, demonstrating the model's ability to distinguish this difficult category. Furthermore, the progressive improvements from fold 2 to fold 5 suggest effective handling of class imbalance.

The consistent diagonal dominance in all confusion matrices, combined with the strong quantitative metrics, provides robust evidence for the model's reliability. These results confirm that our approach successfully addresses key challenges in dynamic facial expression recognition, including class imbalance and discrimination of subtle expression differences, while maintaining robustness across varying data distributions.

## G  VISUALIZATIONS ABOUT HARD REGION DISCOVERY

In this section, we qualitatively validate the effectiveness of the proposed Hard Region Discovery (HRD) mechanism in enhancing DFER. As illustrated in Figure 7, we visualize three key stages of the HRD process: the original input frames, the token-wise importance scores $P$ predicted by HRD, and the final binary masking decisions. These visualizations demonstrate that HRD successfully identifies and prioritizes semantically informative regions across temporal frames.

In prior research, scholars have established the critical role of the eyes and mouth in facial expression recognition. Through eye-tracking analyses of healthy participants' gaze behaviors during the observation of diverse facial expressions, it was revealed that initial fixations were predominantly directed toward either the eyes or the mouth across all emotional categories. Specifically, happy expressions were characterized by longer fixations on the mouth region; in contrast, sad and angry expressions elicited more frequent and prolonged fixations on the eyes. For fearful and neutral expressions, attention allocation to the eyes and mouth was comparable. These findings underscore the pivotal function of the eyes and mouth in emotional decoding, with emotion-specific gaze patterns reflecting a focus on the most diagnostically relevant facial regions for each emotion—thereby confirming their indispensable significance in facial expression recognition.

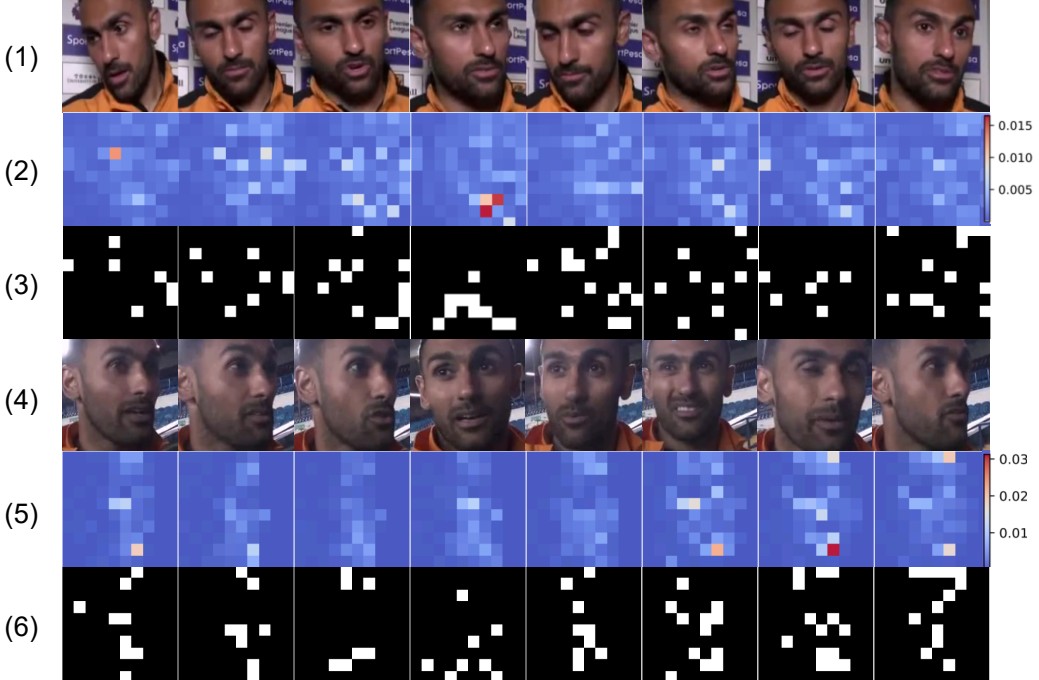

Figure 7: Visualization of the Hard Region Discovery (HRD) process. For the two examples, rows (1) and (4) show input frames, rows (2) and (5) present token-wise importance scores predicted by HRD, and rows (3) and (6) indicate the binary token masking. HRD dynamically selects informative facial regions and suppresses redundant areas across frames.

As established, the eyes and mouth constitute critical regions for facial expression recognition. The visual results in Fig. 7 show that HRD consistently assigns higher probabilities to regions of high emotional salience, such as the eyes, mouth, and eyebrows—areas that are critical for capturing facial expressions—while de-emphasizing low-information or background patches. Notably, the selection patterns adapt across different frames within the same video, reflecting HRD's ability to dynamically modulate token selection based on the frame-wise intensity and semantic complexity of expressions. For instance, during emotionally expressive frames or transitions, HRD increases the number of selected visible tokens (e.g., the mouth corner in Rows (4)-(6) Columns 6-7) to ensure that subtle but significant facial changes are captured by the encoder.

This adaptive discovery mechanism allows the model to concentrate on hard-to-reconstruct tokens, which often correspond to transient or fine-grained facial movements that uniform masking strategies tend to overlook. By selectively exposing these challenging tokens to the encoder, HRD enhances the model's capacity to learn discriminative representations.

In summary, the qualitative results affirm the core advantages of HRD: it facilitates targeted information extraction by adaptively focusing on the most expressive and structurally relevant regions; it provides a principled trade-off between accuracy and efficiency; and it enhances the model's

robustness to occlusions, background clutter, and temporal variation. These properties make HRD a powerful and generalizable mechanism for DFER tasks.

## H  VISUALIZATIONS ABOUT HARD SAMPLE MINING

To well illustrate the sample-level behavior of our Hard Sample Mining (HSM) strategy, we visualize eight representative training samples in Figure 8. Each sample corresponds to a single training instance involved in the loss computation defined by $L_{\text{HSM}} = \sum_i w_i L_{R_i}$, where $L_{R_i}$ denotes the original reconstruction loss of the $i$-th sample, and $w_i$ is the importance weight assigned by HSM.

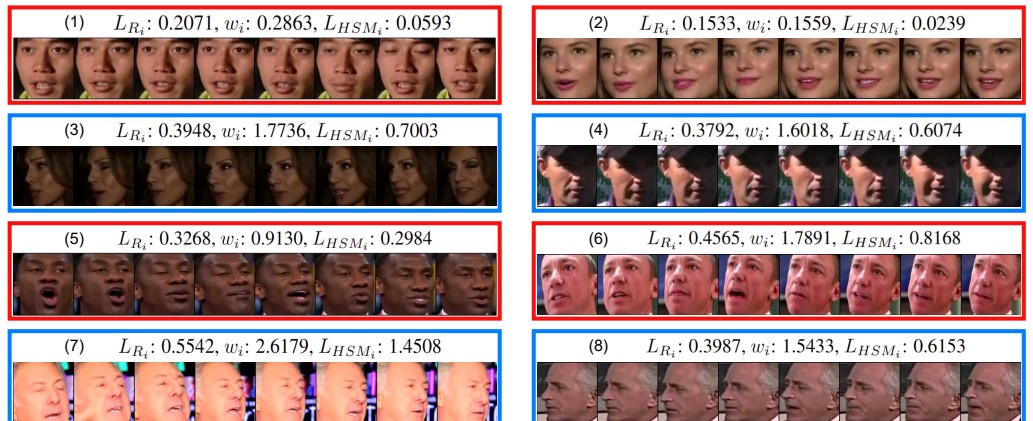

Figure 8: Sample-level visualization under Hard Sample Mining (HSM). Each Square box shows the raw frames of a training sample. Captions indicate the original reconstruction loss $L_{R_i}$, the HSM-assigned weight $w_i$, and the resulting weighted $L_{HSM_i}$ where $i$ denotes the $i-th$ sample. The examples demonstrate that HSM prioritizes semantically informative and challenging samples during training.

In the visualization, only the raw input frames are shown. The title of each subfigure summarizes the key quantities involved in the loss computation: the original loss $L_{R_i}$, the adaptive weight $w_i$, and the resulting weighted $L_{HSM_i}$. These three values directly reflect how HSM modulates each sample's contribution to the total training objective.

The selected samples span a diverse range of difficulties. Samples with high weighted losses typically display significant head movements, exaggerated facial expressions, or challenging lighting and occlusion conditions. In contrast, samples with low weighted losses tend to show small facial changes and clean backgrounds.

We observe that HSM adaptively assigns higher weights to semantically rich or challenging samples—those that are likely to contain more informative content for representation learning. This sample-specific weighting avoids the need for hand-crafted heuristics or hard mining thresholds, and instead leverages reconstruction difficulty as a feedback signal. Overall, this qualitative evidence supports the effectiveness of HSM in emphasizing complex and dynamic facial expression patterns, thereby enhancing the model's robustness and generalization capability.

## I  ABLATION STUDY ON THE IMPORTANCE OF JOINT APPEARANCE-MOTION CONSTRUCTION

To evaluate the individual and combined contributions of appearance and motion modalities in dynamic facial expression recognition, we conduct a comparative ablation study on DFEW fold1. As summarized in Table 9, the joint modeling of both modalities yields the best overall performance, achieving 64.49% UAR and 75.74% WAR, outperforming the appearance-only (63.68% UAR, 75.40% WAR) and motion-only (61.70% UAR, 75.01% WAR) configurations, demonstrating that combining static and dynamic cues mitigates individual limitations. Notably, "Disgust" accuracy

Table 9: Performance comparison of single-modality and joint modeling approaches. Per-class accuracy is reported in the order: [Happy, Sad, Neutral, Anger, Surprise, Disgust, Fear].

| Appearance | Motion | UAR (%) | WAR (%) | Per-class Accuracy (%) |
|:---:|:---:|:---:|:---:|:---:|
| ✓ | - | 63.68 | 75.40 | [75.20, 72.45, 95.30, 33.70, 75.09, 17.24, 76.78] |
| - | ✓ | 61.70 | 75.01 | [74.93, 71.09, 96.52, 36.46, 74.72, 3.45, 74.71] |
| ✓ | ✓ | **64.49** | **75.74** | [74.93, 72.11, 95.91, 34.81, 74.34, 20.69, 78.62] |

improves by 17.24% over motion-only (likely due to appearance's structural cues) and 3.45% over appearance-only (where motion captures subtle movements). This synergy is further evidenced by "Fear" (joint: 78.62% vs. motion-only: 74.71%), highlighting motion's role in transient expressions.

Appearance and motion play asymmetric roles, not equal importance. Appearance serves as a foundation: the appearance-only model outperforms motion-only by 1.98% in UAR (63.68% vs. 61.70%) and leads in "Sad" (72.45% vs. 71.09%) and "Surprise" (75.09% vs. 74.72%), establishing baseline accuracy via static features. Motion adds unique value for temporal-dependent expressions: it boosts "Anger" accuracy (36.46% vs. 33.70% for appearance-only) and, when combined with appearance, mitigates motion-only's poor "Disgust" performance. Thus, appearance provides a stable foundation, motion enhances dynamic sensitivity, and their joint modeling is essential for capturing dynamic facial expressions.

The training dynamics reveal insightful patterns about the interaction between the two modalities. To examine the joint optimization behavior of our dual-branch architecture, we analyze the training dynamics across three key metrics: the appearance branch loss ($\mathcal{L}_1$), motion branch loss ($\mathcal{L}_2$), and their combined reconstruction loss ($\mathcal{L}_R=\mathcal{L}_1 + \mathcal{L}_2$). As shown in Table 10, both reconstruction losses decrease steadily but exhibit distinct convergence behaviors.

Table 10: Evolution of appearance ($\mathcal{L}_1$), motion ($\mathcal{L}_2$), and total ($\mathcal{L}_R$) reconstruction losses during training.

| Epoch | $\mathcal{L}_1$ Mean | $\mathcal{L}_1$ Max | $\mathcal{L}_1$ Min | $\mathcal{L}_2$ Mean | $\mathcal{L}_2$ Max | $\mathcal{L}_2$ Min | $\mathcal{L}_R$ Mean | $\mathcal{L}_R$ Max |
|:---:|:---:|:---:|:---:|:---:|:---:|:---:|:---:|:---:|
| 0 | 1.356 | 2.442 | 0.338 | 1.178 | 4.420 | 0.342 | 1.267 | 1.802 |
| 10 | 0.473 | 3.228 | 0.0004 | 0.520 | 7.275 | 0.0001 | 0.497 | 0.801 |
| 20 | 0.431 | 3.883 | 0.0025 | 0.486 | 7.674 | 0.0003 | 0.459 | 1.066 |
| 30 | 0.428 | 3.047 | 0.0022 | 0.450 | 6.626 | 0.0001 | 0.439 | 0.879 |
| 40 | 0.401 | 3.457 | 0.0022 | 0.437 | 9.114 | 0.0003 | 0.419 | 0.717 |
| 50 | 0.386 | 3.347 | 0.0020 | 0.403 | 6.103 | 0.0001 | 0.395 | 0.881 |
| 60 | 0.321 | 2.631 | 0.0017 | 0.386 | 6.210 | 0.0003 | 0.354 | 0.752 |
| 70 | 0.314 | 3.475 | 0.0032 | 0.388 | 5.728 | 0.0001 | 0.351 | 0.666 |
| 80 | 0.320 | 3.584 | 0.0028 | 0.380 | 6.379 | 0.0002 | 0.350 | 0.541 |

Specifically, the appearance loss $\mathcal{L}_1$ shows rapid initial decline, dropping from 1.356 to 0.473 by epoch 10 (65.1% reduction), then gradually converging to 0.320 by epoch 80. In contrast, the motion loss $\mathcal{L}_2$ demonstrates slower but more consistent improvement, decreasing from 1.178 to 0.380 over the same period. Notably, between epochs 30-60, $\mathcal{L}_1$ enters a stable phase (0.428 to 0.321) while $\mathcal{L}_2$ continues its steady descent (0.450 to 0.386), suggesting that motion representations require longer training to mature.

The total loss $\mathcal{L}_R$ mirrors this coordinated optimization, decreasing smoothly from 1.267 to 0.350 without significant fluctuations. The maximum loss values for both modalities show occasional spikes throughout training (e.g., $\mathcal{L}_2$ max reaches 9.114 at epoch 40), yet the mean losses maintain consistent downward trajectories, demonstrating the robustness of our multi-task learning framework against challenging samples.

These results confirm that appearance and motion play complementary rather than redundant roles. Appearance provides strong structural priors that facilitate rapid early learning, while motion contributes fine-grained temporal patterns that require longer-term optimization. Their joint integration enables comprehensive expression understanding that surpasses single-modality approaches, particularly for dynamic expressions that evolve over time.

# J  GENERALIZATION ANALYSIS ACROSS DEMOGRAPHIC SUBGROUPS

To comprehensively address the generalization capability of our method, particularly concerning potential biases across different demographics, we conduct a detailed subgroup analysis on DFEW Fold 1. we employ the DeepFace framework to annotate each sample with predicted age and race attributes. The dataset is subsequently partitioned into age subgroups ([0-24], [25-35], [36-45], [46-100]) and racial subgroups (Asian, Black, Indian, Latino/Hispanic, Middle Eastern, White). Performance metrics including Weighted Average Recall (WAR), Unweighted Average Recall (UAR), and per-class accuracy are evaluated within each subgroup. The complete results are presented in Table 11.

Table 11: Generalization performance analysis across demographic subgroups on DFEW Fold 1 test set (2,341 samples). Per-class accuracy is reported in the order: [Happy, Sad, Neutral, Anger, Surprise, Disgust, Fear].

| Group | Subgroup | #Samples | WAR | UAR | Per-class Accuracy |
|---|---|---|---|---|---|
| Age | 0-24 | 23 | 60.87 | 61.11 | [100.00,50.00,50.00,50.00,66.67,50.00] |
| | 25-35 | 1537 | 60.57 | 49.86 | [76.35,74.71,69.37,36.40,66.18,8.70,17.29] |
| | 36-45 | 736 | 61.82 | 50.74 | [80.42,73.45,75.14,34.48,65.00,0.00,26.67] |
| | 46-100 | 45 | 60.00 | 37.50 | [62.50,33.33,94.44,22.22,50.00,0.00,0.00] |
| Ethnicity | Asian | 895 | 58.88 | 49.02 | [74.60,75.00,69.77,33.75,61.11,7.14,21.79] |
| | Black | 104 | 59.62 | 54.00 | [72.22,64.29,80.00,44.00,62.50,25.00,30.00] |
| | Indian | 11 | 81.82 | 75.00 | [100.00,100.00,100.00,50.00,100.00,0.00] |
| | Latino/Hispanic | 129 | 62.02 | 49.18 | [81.48,60.87,59.38,60.00,71.43,0.00,11.11] |
| | Middle Eastern | 124 | 64.52 | 58.65 | [82.35,70.59,87.80,41.18,60.00,10.00] |
| | White | 1078 | 62.06 | 49.58 | [79.15,74.85,71.37,32.66,69.17,0.00,19.18] |

Three key observations emerge from the subgroup analysis: (1) **Consistent WAR**: The Weighted Average Recall remains stable across all age groups (60.00%–61.82%) and ethnicities (58.88%–64.52%), indicating robust overall classification performance regardless of demographic attributes. (2) **UAR Variability**: The Unweighted Average Recall shows greater variability, particularly in smaller subgroups (e.g., 37.50% for age 46-100, 75.00% for Indian), which can be attributed to the increased sensitivity of UAR to class imbalance in limited-sample settings. (3) **Per-class Patterns**: The model maintains high accuracy for universally distinguishable expressions such as "Happy" and "Neutral" across all subgroups, while challenges remain for "Disgust" and "Fear" due to their inherent subtlety and lower prevalence in training data.

Notably, the model demonstrates particularly strong performance in the Black subgroup (54.00% UAR) and Middle Eastern subgroup (58.65% UAR), achieving the highest "Anger" recognition accuracy among all ethnicities (44.00% and 41.18%, respectively). This suggests that the learned representations effectively capture expression characteristics that transcend racial phenotypes.

The results provide empirical evidence that our method achieves commendable generalization across diverse demographic groups. The consistent performance stems from the model's focus on learning dynamic facial muscle movements rather than static demographic features. While the current evaluation on established benchmarks (DFEW and FERV39K) demonstrates strong cross-population robustness, we acknowledge the value of further validation on more extensive and demographically balanced datasets as an important future direction.

# K  ANALYSIS OF ACTIVATION FUNCTION SELECTION

Table 12: Performance comparison of different activation functions in the HSM module on DFEW fold1.

| Activation Function | WAR (%) | UAR (%) |
|---|---|---|
| Softplus | **75.74** | **64.49** |
| ELU | 75.27 | 63.30 |
| Swish | 75.22 | 63.02 |
| ReLU | 75.10 | 62.85 |

Table 13: Effects of different random seeds. Per-class accuracy is reported in the order: [Happy, Sad, Neutral, Anger, Surprise, Disgust, Fear].

| Value | UAR (%) | WAR (%) | Per-class Accuracy (%) |
|---|---|---|---|
| Original (0) | 64.49 | 75.74 | [95.91, 74.93, 74.34, 78.62, 72.11, 20.69, 34.81] |
| 1 | 64.52 | 75.78 | [96.32, 78.36, 72.28, 76.78, 71.43, 17.24, 39.23] |
| 42 | 64.48 | 75.82 | [95.91, 74.41, 77.90, 77.01, 67.35, 20.69, 38.12] |
| 2025 | 64.52 | 75.74 | [95.91, 75.20, 75.84, 76.32, 71.77, 20.69, 35.91] |

Within the Hard Sample Mining (HSM) framework, the activation function must convert sample loss deviations into non-negative importance weights. We evaluate four candidate functions on DFEW fold 1, with results summarized in Table 12.

Softplus achieves superior performance due to its ideal properties for this task: (1) It guarantees smooth, non-negative outputs essential for stable weight assignment; (2) It provides gentle non-linearity that preserves relative loss relationships without aggressive saturation; (3) It maintains differentiability across all inputs, enabling effective gradient flow during optimization. While ReLU's hard zero-clipping may discard valuable negative deviation information, and ELU/Swish introduce unnecessary complexity, Softplus optimally balances mathematical suitability with empirical performance.

## L EFFECTS OF DIFFERENT RANDOM SEEDS

To further validate the robustness of our results, we supplemented this with additional fine-tuning experiments using different random seeds (1, 42, 2025) on DFEW fold 1, alongside our original seed (0). The results demonstrate minimal fluctuation across all seeds: UAR varies by only 0.04% (max 64.52% vs. min 64.48%); WAR fluctuates by 0.08% (max 75.82% vs. min 75.74%); Both metrics show variances <0.0001. This consistency confirms that our method's performance gains are not accidental but stem from inherent improvements in the model. That said, the negligible variance in our multi-seed results—coupled with the stable improvement trend over the reported baseline performance—strongly supports the reliability of our conclusions.

