# OpenReview forum: "HI2M: Hard Inter- and Intra-Sample Masking for Dynamic Facial Expression Recognition"
_ICLR.cc/2026/Conference — ICLR 2026 Conference Withdrawn Submission_

### Official Review · Reviewer_MpC4 · 2025-10-30

**Soundness:** 2
**Presentation:** 3
**Contribution:** 2
**Rating:** 4
**Confidence:** 3

**Summary:**

This paper proposes HI2M, a self-supervised framework for Dynamic Facial Expression Recognition (DFER) that enhances Masked Autoencoder (MAE) pre-training through a hierarchical, difficulty-aware strategy. The framework consists of Hard Sample Mining (HSM), which re-weights samples based on inter-sample reconstruction difficulty, and Hard Region Discovery (HRD), which employs reinforcement learning to adaptively find the most informative intra-sample spatiotemporal regions. Extensive experiments on DFEW and FERV39K demonstrate that HI2M achieves state-of-the-art performance, validating its dual-level hard-mining design. Overall, the work presents a promising approach for improving video representation learning by focusing on semantically rich samples and regions. However, certain aspects such as computational cost and RL stability warrant further discussion.

**Strengths:**

1. The hierarchical masking strategy is novel and well-motivated, simultaneously addressing inter-sample and intra-sample importance.

2. HSM provides a simple yet effective batch-wise re-weighting mechanism, while HRD introduces an innovative reinforcement learning-based adaptive masking policy.

3. The empirical evaluation is comprehensive, including extensive ablation studies that clearly show the contributions of HSM and HRD individually and synergistically.

4. Quantitative results demonstrate state-of-the-art performance on challenging DFER benchmarks, supported by qualitative visualizations illustrating attention to salient facial dynamics.

**Weaknesses:**

1. The RL-based HRD module introduces additional complexity and computational overhead, yet the paper lacks quantitative analysis of training time or FLOPs compared to baseline MAE models; including a brief discussion on computational cost and scalability would enhance the paper's practical clarity and applicability.

2. The stability and sensitivity of the RL training process are not analyzed—given the known instability of policy gradient methods, a discussion on hyperparameter sensitivity (e.g., policy learning rate α) would improve reproducibility.

3. HSM's reliance on batch statistics for sample weighting may make performance dependent on batch size, which is not examined. While baseline comparisons are thorough, evaluation against other self-supervised hard-mining techniques could better contextualize the novelty of HI2M.

4. The HRD reward function, based on reconstruction loss, could inadvertently prioritize noisy or unpredictable regions rather than purely informative ones; discussing potential mitigation strategies would strengthen the work.

5. Additionally, the term "hard token" appears in Figure 2 but is not formally defined in the main text—it seems to correspond to the "visible tokens" selected by HRD according to token difficulty. Clarifying this terminology would improve consistency and reader comprehension.

**Questions:**

Please see the Weakness part and respond to the corresponding concerns.

---

### Official Review · Reviewer_nX9f · 2025-11-01

**Soundness:** 3
**Presentation:** 3
**Contribution:** 2
**Rating:** 4
**Confidence:** 5

**Summary:**

This paper proposes HI2M, a self-supervised framework for dynamic facial expression recognition (DFER) that addresses both inter-sample difficulty and intra-sample region importance during masked reconstruction. The method introduces two components: Hard Sample Mining (HSM), which prioritises difficult samples based on reconstruction loss, and Hard Region Discovery (HRD), which selectively masks informative spatiotemporal regions to focus learning on expression-relevant dynamics. By adapting masking at both the sample and region level, the method aims to improve feature quality for downstream recognition. Experiments on two benchmark datasets demonstrate state-of-the-art performance compared to existing self-supervised DFER approaches.

**Strengths:**

1. The paper clearly articulates two challenges in self-supervised DFER—inter-sample difficulty and intra-sample region informativeness—and attempts to address them in a unified framework.
2. The method achieves state-of-the-art results on two benchmark DFER datasets, demonstrating its empirical effectiveness in downstream expression recognition tasks.

**Weaknesses:**

1. The motivation is somewhat overstated: the claim that “equal treatment of all samples and regions is a limitation of DFER” only applies to masked autoencoder–based approaches, not to the broader DFER literature, which includes many supervised and non-masked paradigms.
2. The rationale behind using token-wise reconstruction loss as a measure of region difficulty is not well justified, especially given that faces are spatially aligned and simpler alternatives (e.g., landmark-based region selection) may work as effectively.
3. The paper assumes that higher reconstruction loss correlates with higher temporal dynamics, but no qualitative or quantitative evidence is provided to confirm this assumption; spatial texture complexity may also influence the loss.
4. For inter-sample difficulty, the choice of reconstruction loss as the weighting metric lacks comparison to simpler baselines (e.g., temporal frame-difference magnitude), which would help validate the design choice.
5. The performance gains from the two core modules (HRD and HSM) are relatively small (0.49% and 0.79%), and no statistical robustness (mean/std over multiple runs) is reported, making it difficult to assess whether the improvements are significant or within random variance.

**Questions:**

1. Can the authors provide evidence (visualisations or statistics) that higher reconstruction loss indeed corresponds to rapid temporal variation rather than spatial complexity or noise?
2. Have multiple seeds or standard deviations been reported to verify that the observed accuracy gains are not within random fluctuation?
3. Were simpler alternatives (e.g., landmark-based region masking, frame–difference–based sample difficulty) tested, and if so, how do they compare to HRD and HSM?

---

### Official Review · Reviewer_MbTX · 2025-11-01

**Soundness:** 3
**Presentation:** 3
**Contribution:** 3
**Rating:** 6
**Confidence:** 3

**Summary:**

This paper proposes Hard Inter- and Intra-sample Masking (HI2M) to address the weakness of existing MAE-based approaches in self-supervised learning for Dynamic Facial Expression Recognition (DFER), which uniformly treat all samples and regions, leading to insufficient attention to informative samples and regions. HI2M integrates Hard Sample Mining (HSM), which applies adaptive weighting based on inter-sample difficulty differences, and Hard Region Discovery (HRD), which selectively masks informative spatiotemporal regions within samples, thereby achieving reconstruction learning with emphasis on hard examples.

**Strengths:**

- The paper proposes an idea of difficulty-adaptive learning for self-supervised learning in DFER.
- Based on this idea, the paper introduces novel modules: Hard Sample Mining (HSM) and Hard Region Discovery (HRD).
- The method demonstrates high effectiveness on DFEW and FERV39k datasets.

**Weaknesses:**

- In the confusion matrix analysis in Section 4.5, the logic for claiming "(1) verifying consistent attention to discriminative facial regions across different data distributions" based on this analysis is unclear. This claim may require analyses similar to those described in Appendices G and H.
- The method for determining hyperparameters is unclear. In the DFER field, cross-validation is commonly used for evaluation; however, unfortunately, the erroneous practice of determining hyperparameters on the test split without preparing a separate validation split is prevalent. To dispel concerns that this paper may be using such an incorrect method, please clearly explain how the hyperparameters were determined.
- (Minor comment) Non-variable subscripts such as HSM in L_HSM should be typeset in roman font. The "log" in Equation (7) should also be in roman font.

**Questions:**

- The accuracy of Random init. in Table 3 seems excessively low. Could you please explain the experimental setup and reasons for this result?

---

### Official Review · Reviewer_TXgK · 2025-11-02

**Soundness:** 3
**Presentation:** 3
**Contribution:** 2
**Rating:** 4
**Confidence:** 5

**Summary:**

To address the issue of equal treatment of all samples and regions in dynamic facial expression recognition,this paper proposes the HI2M framework, which incorporates Hard Sample Mining (HSM) and Hard Region Discovery (HRD). HSM dynamically prioritizes difficult-to-reconstruct samples, while HRD adaptively focuses on informative facial regions using reinforcement learning. Experimental results on benchmark datasets (DFEW and FERV39K) demonstrate that HI2M outperforms existing methods.

**Strengths:**

1. The proposed method concurrently addresses hardnessaware masking at both sample and region granularities.
2. This paper is easy to follow.

**Weaknesses:**

1. Marginal Improvements: According to Table 1, the proposed method shows only marginal improvements in performance, suggesting that the introduced mechanisms for selecting hard samples and regions may have limited effectiveness.

2. Handling of Dynamic Videos by HRD: The paper focuses on dynamic facial expression recognition, yet the explanation of how the Hard Region Discovery (HRD) module specifically addresses the temporal dynamics in videos is insufficient. For instance, it is unclear whether HRD focuses solely on spatial regions or incorporates temporal-spatial regions. The current region selection mechanism appears applicable to static images as well, and it is not clear what unique advantages HRD provides in handling dynamic video sequences. Additionally, it is important to clarify whether HRD, like HSM, operates on both appearance and motion streams for region selection.

3. Non-differentiability of Categorical Function: In Eq. (6), the visible tokens are selected via a Categorical function, which is a non-differentiable operation. This raises concerns about whether this operation can be effectively trained via backpropagation. A detailed discussion of the training strategy and how this non-differentiable selection is handled during optimization would be beneficial.

4. Missing Details in Experiments:

Pre-training Corpus: In Section 4.3, the proposed method is compared with existing MAE-based approaches [Sun et al. (2023; 2024); Wu et al. (2025)], all of which utilize unlabeled data. However, the paper does not clarify which corpus was used for pre-training in these methods. Is it the same corpus as the one used in this paper? Clarifying this point would provide a better comparison.

Ablation Study: Table 2 presents an ablation study of different components, but it is unclear which dataset was used. Since HSM and HRD are core components of the proposed method, it would be more informative to conduct the ablation study on both the DFEW and FERV39K datasets to evaluate the generalizability of the method.

**Questions:**

1. The improvement is marginal.
2. Details of the methods design and training strategy.
3. Miss details in the experiments.

---

### Note · Authors · 2025-11-20

I have read and agree with the venue's withdrawal policy on behalf of myself and my co-authors.